# Genetic underpinnings of regional adiposity distribution in African Americans: Assessments from the Jackson Heart Study

**Mohammad Y. Anwar**[1]*, **Laura M. Raffield**[2], **Leslie A. Lange**[3], **Adolfo Correa**[4], **Kira C. Taylor**[1]

**1** School of Public Health & Information Sciences, The University of Louisville, Louisville, KY, United States of America, **2** Department of Genetics, University of North Carolina, Chapel Hill, NC, United States of America, **3** Division of Biomedical Informatics and Personalized Medicine, University of Colorado School of Medicine, Aurora, Colorado, United States of America, **4** Jackson Heart Study, Department of Medicine, University of Mississippi Medical Center, Jackson, Mississippi, United States of America

☯ These authors contributed equally to this work.

* m0anwa02@louisville.edu

## Abstract

### Background

African ancestry individuals with comparable overall anthropometric measures to Europeans have lower abdominal adiposity. To explore the genetic underpinning of different adiposity patterns, we investigated whether genetic risk scores for well-studied adiposity phenotypes like body mass index (BMI) and waist circumference (WC) also predict other, less commonly measured adiposity measures in 2420 African American individuals from the Jackson Heart Study.

### Methods

Polygenic risk scores (PRS) were calculated using GWAS-significant variants extracted from published studies mostly representing European ancestry populations for BMI, waist-hip ratio (WHR) adjusted for BMI (WHR$_{BMIadj}$), waist circumference adjusted for BMI (WC$_{BMIadj}$), and body fat percentage (BF%). Associations between each PRS and adiposity measures including BF%, subcutaneous adiposity tissue (SAT), visceral adiposity tissue (VAT) and VAT:SAT ratio (VSR) were examined using multivariable linear regression, with or without BMI adjustment.

### Results

In non-BMI adjusted models, all phenotype-PRS were found to be positive predictors of BF%, SAT and VAT. WHR-PRS was a positive predictor of VSR, but BF% and BMI-PRS were negative predictors of VSR. After adjusting for BMI, WHR-PRS remained a positive predictor of BF%, VAT and VSR but not SAT. WC-PRS was a positive predictor of SAT and VAT; BF%-PRS was a positive predictor of BF% and SAT only.

**Data Availability Statement:** The data underlying the findings include potentially identifying participant information and cannot be made

publicly available due to ethical/legal restrictions. However, data, including statistical code, from this manuscript are available to researchers who meet the criteria for access to confidential data. Access to JHS data is available through (https://www.jacksonheartstudy.org/Research/Study-Data/Data-Access). All JHS genetic data is available through dbGaP at phs000286.v6.p2. Additionally, much of the JHS data is available through BioLINCC (https://biolincc.nhlbi.nih.gov/studies/jhs/).

**Funding:** Acknowledgements The Jackson Heart Study (JHS) is supported and conducted in collaboration with Jackson State University (HHSN268201800013I), Tougaloo College (HHSN268201800014I), the Mississippi State Department of Health (HHSN268201800015I) and the University of Mississippi Medical Center (HHSN268201800010I, HHSN268201800011I and HHSN268201800012I) contracts from the National Heart, Lung, and Blood Institute (NHLBI) and the National Institute on Minority Health and Health Disparities (NIMHD). The authors also wish to thank the staffs and participants of the JHS. LMR was supported by T32 HL129982. The funders for LMR had no role in study design, data collection and analysis, decision to publish, or preparation of the manuscript.

**Competing interests:** The authors have declared that no competing interests exist.

## Conclusion

These analyses suggest that genetically driven increases in BF% strongly associate with subcutaneous rather than visceral adiposity and BF% is strongly associated with BMI but not central adiposity-associated genetic variants. How common genetic variants may contribute to observed differences in adiposity patterns between African and European ancestry individuals requires further study.

## Introduction

Despite the wide adoption of body mass index (BMI), a measure that correlates well with numerous health risk factors [1], there are limitations to this metric [2]; notably, it does not differentiate between variation in fat and lean mass. This can lead to an imprecise categorization of obesity [3], and hence misleading inferences for cardiometabolic outcomes [4]. This is a major drawback given that obesity-health outcome associations are differentially mediated by bodily composition, with adipose tissues more prominently linked with adverse outcomes [5]. Total adiposity [6], regional fat distribution [7], and especially visceral adiposity [8] are significant risk factors for cardiovascular diseases.

Since body fat is linearly associated with BMI in sedentary populations [9], a proportion of genetic variants associated with overall body mass expectedly overlap with loci linked to body fat percentage [10]. However, this BMI-body fat mass link is known to exhibit phenotypic variability across ethnicities [11], which extends to regional distribution of fat tissue as well: in African ancestry (AA) individuals with comparable BMI metrics to those with European ancestry (EA), the proportion of visceral adiposity is lower [12, 13]. Paradoxically, the prevalence of cardiometabolic diseases in AA, compared to EA with similar BMI, is higher [14]; genetics may also play a significant role in this differential adiposity distribution and merits a thorough examination.

Despite some evidence for AA-specific variants associated with BMI [15], most reported BMI-associated SNPs in AAs are variants first reported in EA individuals [16], and European-derived genetic risk scores have been found to be predictive of BMI variation in AA populations as well [17]. But evidence for generalization of variants associated with other adiposity traits to AAs, including measures of central obesity, is limited [18], and no study has examined replicability of body fat percentage-associated variants identified in EA populations to AA. Since genomic loci associated with body fat percentage are suggested to be more closely aligned with multiple cardiometabolic disease risks than BMI-associated variants [19], assessing if genetic variants associated with adiposity patterns in EA—particularly body fat percentage (BF%)—can be extrapolated to AA individuals is an important question.

The objective of this this study was to assess the utility of known variants associated with anthropometric and adiposity measures discovered in predominantly EA populations to predict BF%, SAT, VAT, and VSR among AA individuals from Jackson Heart Study (JHS). We also estimated the association of variants with evidence of directional replicability and nominal significance in JHS for their originally reported adiposity measure to other adiposity measures, allowing us to examine the relationships between adiposity measures.

## Methods

### Study population

The JHS recruited 5306 African American residents living in the Jackson, Mississippi, metropolitan area of Hinds, Madison, and Rankin Counties. Participants were recruited to

participate in the study from four pools: random sampling (17% of participants), volunteers (30%), participants in the Atherosclerosis Risk in Communities (ARIC) study (31%), and secondary family members (22%). The age at enrollment for the unrelated cohort was 35–84 years; the family cohort included related individuals >21 years old. Extensive phenotypic data was collected during a baseline examination (September 2000 –March 2004), and two follow-up examinations (October 2005 –December 2008, and February 2009 –January 2013). A third follow-up examination is in progress. Annual follow-up interviews and cohort surveillance for cardiovascular events and mortality are also ongoing.

For BMI and WC genome-wide association tests, we used phenotypic observations from visit 1 to maximize sample size (N = 3020); for other traits, we used phenotypic measures from visit 2 for the same purpose (N = 2554). However, for polygenic risk score regression analyses, we exclusively used phenotypic observations from visit 2. In our analyses, 10 participants were excluded due to pregnancy during the visit 2 examination, and 124 were excluded for missing or biologically implausible values. The total sample size used was n = 2420 individuals.

The study protocol was approved by the participating JHS institutions including the Tougaloo College, the Jackson State University, and the University of Mississippi Medical Center IRB, under IRB number- UMMC IRB File#1998–6004. All JHS participants gave written informed consent. The IRB registration for UMMC is #00000061, DHHS FWA #00003630.

## Genotyping

Genotyping for single nucleotide polymophisms (SNPs) was performed with the Affymetrix 6.0 SNP Array (Affymetrix, Santa Clara, Calif). Outliers based on principal components, sample swaps, duplicates, and one of each pair of monozygotic twins were excluded. Samples with a mismatch between pedigree vs genetic sex were also removed. Variants with a minor allele frequency $\geq$ 1%, a call rate $\geq$ 90%, and a Hardy Weinberg equilibrium (HWE) p-value >10–6 (n = 832,508 variants) were used for imputation to 1000 Genomes Project population SNP reference panel (Phase 3, Version 5), using Minimac3 on the Michigan Imputation Server. Only SNPs with an imputation quality $r^2$>0.9 were selected for polygenic risk score analyses.

## Polygenic risk scores

The use of polygenic risk scores (PRS) to predict complex traits [20], closely related phenotypes [21], and the same phenotype across different populations [22] has been validated. Both weighted and unweighted methods have been employed for estimation of PRS [23]. Weighted risk scores account for each variant's effect size on the phenotype. Unweighted PRS assume that all variants have equal effect on the trait; this assumption is almost always violated, as in genome-wide association studies (GWAS), some variants have much larger effect sizes than others. In contrast, use of a weighted PRS method in a non-EA population may also introduce bias because nearly all known variants were identified in predominantly EA populations, and studies suggest dissimilar effect sizes for the same SNPs across ancestries, likely due to differential LD with true causal variants [24]. Although the weighted method can lead to reduced mean square error for prediction in some cases [25], the main applications for polygenic scores, namely association testing and prediction, do not appear to differ substantially between two methods. In addition, an unweighted score is more robust against error in estimating the effect sizes arising from limited samples, "winner's curse bias" [26], and confounding by demographic structure [27]. Therefore, we used an unweighted PRS approach in the AA population studied here.

At each locus (SNP), participants were assigned a dosage value between 0 and 2 inclusive, based on the estimated number and frequency of phenotype-increasing alleles under an

additive genetic effect model. The PRS value for each individual reflects the summation of risk alleles across all selected loci.

## SNP risk sets

To construct SNP sets used for PRS calculation, we utilized both the European Bioinformatic Institute GWAS repository (*ebi.ac.uk/gwas, accessed December 2020*), and PubMed for extraction of SNPs linked to anthropometric measures including body mass index (BMI), waist to hip ratio adjusted for BMI (WHR$_{BMIadj}$), waist circumference adjusted for BMI (WC$_{BMIadj}$), and body fat percentage (BF%). Only SNPs reported at GWAS significance level (p≤5.00 ×10$^{-8}$) were selected. Most variants were discovered in EA-only studies, but some are from large multi-ethnic GWAS meta-analyses or AA specific analyses. PRS derived from only AA-specific variants for target phenotypes were judged to be underpowered due to the low number of available variants, as well as PRS calculated from SNPs associated with VAT, SAT and VSR.

We conducted linkage disequilibrium (LD) analysis using LDlink (*ldlink.nci.nih.gov*) with multi-ethnic populations. If a pair or a group of SNPs were in LD with one another (R$^2$≥0.1), we prioritized sentinel variants from larger and more recent studies with lower p-values and selected a single SNP to ensure independence of variants and avoid double counting the same functional locus (site). This set of LD-pruned variants was then used to calculate the PRS.

## Anthropometry measures

WC was measured at the umbilicus level using a non-elastic tape measurer and rounded to the nearest centimeter; hip circumference (HC) was measured at the level of the widest circumference over the greater trochanter. WHR was obtained by dividing WC over HC.

## Adiposity measures

A variety of different techniques for assessment of body composition exist [28]. For overall BF %, bioimpedance is a widely adopted method [29]. Under this technique, BF% is calculated based on the measured resistance of the adipose tissue as the person lays supine with electrodes placed on the arm and/or leg; bare foot-to-foot bioimpedance was conducted using the Tanita Body Fat Monitor (Tanita Corp, Tokyo). BF% was estimated using a programmed algorithm that incorporates bioimpedance readings with a height, weight, age and sex-specific equation and additional adjustment for physical activity levels.

To estimate visceral and subcutaneous adipose volumes, the study employed computed tomography (CT) technique at visit 2, where the heart and lower abdomen regions were scanned with 16-channel mutidetector CT machine (Lightspeed 16 Pro, GE Healthcare, Milwaukee, WI). Abdominal imaging slices covering the lower abdomen from L3 to S1 were used to quantify both VAT and SAT [29], such that 24 adjacent 2-mm thick slices centered on the lumbar disk space at L4 to L5 were used for quantification of both types of adiposity; 12 images before the center of L4 to L5 disk space and 12 images after that space [30].

## Statistical analysis

To facilitate comparison across different phenotypes, we performed inverse-normal transformations prior to analysis. Genome-wide association analyses were completed for target traits. We used EPACTS 3.2.6 [31] to perform GWAS analyses, adjusting for age, sex, and a genetic relationship matrix using the EMMAX test; additionally, BMI was incorporated as covariate in WHR and WC analyses.

We calculated PRS under three separate but complementary scenarios using the following configurations: (a) set of all known loci (LD-pruned) reported at genome-wide significance ($5\times10^{-8}$) in multi-ethnic or European studies regardless of replicability in JHS (principle approach) (S1 Table); (b) the subset of risk loci with evidence of directional replication in the JHS-GWAS results (approach 2); and (c) a more restricted subset of risk loci with evidence for both directional replication as well as nominal statistical significance ($p<5\times10^{-2}$) in JHS results (approach 3, S2 Table). PRS for BMI, $WHR_{BMIadj}$, $WC_{BMIadj}$, and BF% were first tested against their respective phenotypes in JHS to ensure the validity of constructed predictors (S3 Table).

Phenotype-specific PRS obtained under various approaches were then tested for cross-sectional associations with phenotypic measures. Results for PRS obtained under the principle approach (e.g. all known loci) were reported in the manuscript, whereas models that were obtained with PRS under complementary approaches were provided as supplementary material. Both multivariable linear regression and mixed models [32] were employed to investigate the associations between PRS and adiposity measures, with age, sex, and the top 10 ancestry principal components as covariates; additionally, family ID was utilized as random component in mixed models. Both offered similar results; linear results were chosen for simplicity.

Some loci associated with obesity traits and/or fat distributions [33] are known to be sex-specific. Although we performed gender-stratified analysis, given the sex imbalance in this sample (36.9% males), results for the females largely mirrored the combined findings, while the male-only analysis lacked precision; therefore we chose to report the sex-combined analysis.

Finally, to characterize the association between EA-established variants for anthropometric traits with evidence of transferrability to AA (i.e. statistically significant in JHS-GWAS), and type of adiposity (BF%, SAT, VAT, VSR), we constructed heatmap plots of SNPs' effect sizes (betas*100) to investigate if genetic variations underpinning obesity traits are closely correlated with overall body fat change or aligned to specific adiposity patterns. For phenotype and SNP clustering, *Ward's* minimum variance method was used which aims at finding compact, spherical clusters [34].

Statistical analyses including PRS generation and regression models were performed using RStudio (V 1.1.463), and heatmap plots were obtained using the R package "pheatmap" [35].

## Results

### Population characteristics and adiposity measures

Table 1 provides a descriptive distribution of demographic, anthropometric and adiposity traits in the JHS population. Using the standard BMI cutpoints of $\geq25$ and $\geq30$ for overweight and obesity respectively, the majority of participants were either overweight (31.5%) or obese (54.2%). The mean WC and BF% also indicate a high prevalence of excess adiposity.

Spearman correlation coefficients for anthropometric and adiposity measures (S4 Table) illustrated a high degree of correlation of BMI with WC (r = 0.82), but much weaker correlation with WHR (r = 0.16). BMI was also highly correlated with SAT (r = 0.83) and BF% (r = 0.70), but not as strongly with VAT (r = 0.49).

### Association with polygenicc risk scores

In non-BMI adjusted models, all phenotype-specific PRS were found to be positive predictors of BF%; β = 2.4 (p = 2.1 $\times10^{-10}$) for BF%-PRS; β = 1.2 (p = 3.3 $\times10^{-51}$) for BMI-PRS; β = 1.1 (p = 2.2 $\times10^{-19}$) for WC-PRS; and β = 0.7 (p = 1.4 $\times10^{-8}$) for WHR-PRS respectively, where β represents % change in phenotype z-score per increase of 1 trait-increasing allele (Table 2). For the SAT phenotype, like BF%, all phenotype-specific PRS were positive predictors: β = 1.9

**Table 1. Baseline characteristics of the study participants.**

| Variable | Total (N = 2420) |
|---|---|
| **Male**, N. (%) | 892 (36.9%) |
| **AGE (years)**, Mean (SD) | 60.0 (12.4) |
| **BMI (kg/m²)**, Mean (SD), Unit | 32.2 (7.2) |
| **BMI Categories** | |
| Underweight, N. (%) | 26 (1.1%) |
| Normal Weight, N. (%) | 244 (10.1%) |
| Overweight, N. (%) | 762 (31.5%) |
| Obese, N. (%) | 1312 (54.2%) |
| Missing, N. (%) | 76 (3.1%) |
| **Hip Circumference (cm)**, Mean (SD) | 114.7 (14.9) |
| **Waist Circumference (cm)**, Mean (SD) | 102.9(16.2) |
| **Waist/Hip (ratio* 100)**, Mean (SD) | 89.8(8.8) |
| Adiposity Composition | |
| **Fat Mass (kg)**, Mean (SD) | 76.3(33.5) |
| **VAT (cm³)**, Mean (SD), Unit | 839.4(383.1) |
| **SAT (cm³)**, Mean (SD), Unit | 2335.8(1014.7) |
| **Body Fat Mass (%)**, Mean (SD) | 38.2(9.9) |

**VAT**: Visceral Adipose Tissue, **SAT**: Subcutaneous Adipose Tissue.

(p = $1.9 \times 10^{-4}$) for BF%-PRS; β = 1.4 (p = $1.4 \times 10^{-37}$) for BMI-PRS; β = 1.3 (p = $9.8 \times 10^{-15}$) for WC-PRS; and β = 0.6 (p = $5.8 \times 10^{-5}$) for WHR-PRS respectively. For the VAT phenotype, BMI-PRS and WC-PRS were positively associated with similarly close coefficients (e.g. slope) as the SAT phenotype (Table 2), but for WHR-PRS, the coefficient was notably larger (β = 1.2 (p = $3.5 \times 10^{-9}$)); BF%-PRS was positive but not strong predictor of VAT (β = 0.8 (p = $1.4 \times 10^{-1}$)). For the VSR, BF%-PRS was a significant negative predictor (β = -0.9 (p = $5.2 \times 10^{-2}$)) and WHR-PRS was a positive predictor (β = 0.4 (p = $3.8 \times 10^{-3}$)).

In contrast, after adjusting for BMI in the models, only BF% and WHR were found to be positive predictors of BF%, with attenuated effect sizes (β = 1.3 (p = $2.1 \times 10^{-7}$) for BF%-PRS

**Table 2. Associations between phenotype-PRS (columns), and measures of adiposity (rows).** Betas are reported for standardized inverse normalized values, followed by their respective p-values. Nominally statistically significant results (p<$5.00 \times 10^{-2}$) are in bold font.

| Phenotype-PRS/Adiposity trait | BF% β (p-value) (95%CI) | SAT β (p-value) (95%CI) | VAT β (p-value) (95%CI) | VAT: SAT R. β (p-value) (95% CI) | BMI Adjusted |
|---|---|---|---|---|---|
| WHR | **0.7 ($1.4 \times 10^{-8}$) (0.4, 0.9)** | **0.6 ($5.8 \times 10^{-5}$) (0.3, 1.0)** | **1.2 ($3.5 \times 10^{-9}$) (0.8, 1.5)** | **0.4 ($3.8 \times 10^{-3}$) (0.1, 0.7)** | No |
| WC | **1.1 ($2.2 \times 10^{-19}$) (0.9, 1.3)** | **1.3 ($9.8 \times 10^{-15}$) (0.9, 1.6)** | **1.2 ($6.9 \times 10^{-11}$) (0.8, 1.5)** | -0.0 ($6.9 \times 10^{-1}$) (-0.3, 0.3) | |
| BMI | **1.2 ($3.3 \times 10^{-51}$) (1.1, 1.4)** | **1.4 ($1.0 \times 10^{-37}$) (1.2, 1.6)** | **1.1 ($2.0 \times 10^{-20}$) (0.9, 1.4)** | **-0.2 ($2.2 \times 10^{-2}$) (-0.4, 0.0)** | |
| BF% | **2.4 ($2.1 \times 10^{-10}$) (1.7, 3.2)** | **1.9 ($1.9 \times 10^{-4}$) (0.9, 2.9)** | 0.8 ($1.4 \times 10^{-1}$) (-0.3, 2.0) | **-0.9 ($5.2 \times 10^{-2}$) (-1.9, 0.0)** | |
| WHR | **0.2 ($2.6 \times 10^{-3}$) (0.1, 04)** | 0.1 ($5.5 \times 10^{-1}$) (-0.1, 0.2) | **0.7 ($6.7 \times 10^{-7}$) (0.4, 1.0)** | **0.5 ($3.3 \times 10^{-4}$) (0.2, 0.8)** | Yes |
| WC | 0.1 ($3.2 \times 10^{-1}$) (-0.1, 0.2) | **0.2 ($3.0 \times 10^{-2}$) (0.0, 0.4)** | **0.3 ($2.8 \times 10^{-2}$) (0.0, 0.6)** | 0.1 ($3.8 \times 10^{-1}$) (-0.2, 0.4) | |
| BMI | | | | | |
| BF% | **1.3 ($2.1 \times 10^{-7}$) (0.8, 1.8)** | **1.0 ($2.7 \times 10^{-4}$) (0.4, 1.6)** | 0.1 ($9.1 \times 10^{-1}$) (-0.9, 1.0) | -0.8 ($8.2 \times 10^{-2}$) (-1.8, 0.1) | |

**WHR**: Waist to Hip Ratio, **WC**: Waist Circumference, **BF%**: Body Fat Percentage, **SAT**: Subcutaneous Adipose Tissue, **VAT**: Visceral Adipose Tissue, **VAT/SAT R.**: VAT to SAT Ratio, **β**: effect size (% change in z-score per increase in number of risk alleles).

* Associations adjusted for age, sex, first 10 ancestry principle components and BMI. Associations estimated under approach 1 (using all variants in the study).

and β = 0.2 (p = 2.6 ×10$^{-3}$) for WHR-PRS respectively) (Table 2). For the SAT phenotype, BF% and WC-PRS remained positive predictors, but not WHR-PRS. For the VAT phenotype, both WHR and WC-PRS were positively associated, but not BF%-PRS. For the VSR, BF%-PRS was consistent negative predictor and WHR-PRS was positive predictor. Generally, the PRS of BMI and BF% phenotypes explained higher proportions of variance for BF% and SAT than VAT (S4 Table).

Results for PRS obtained under approach 2 and 3 are provided in S5 Table, which were largely reflective of estimates under the principle approach but less precise due to reduced number of SNPs used for the PRS calculation. Associations between WHR and WC-PRS were less consistent in both BMI and non-BMI adjusted models. BF%-PRS, in contrast, was consistently associated with BF% and SAT under both approach 2 and 3 and with/without BMI adjustment.

## Association with individual SNPs

Since PRS were summary statistics of risk alleles and do not illustrate the scale of association of each risk allele, which may vary when compared to other variants, we performed separate multivariate regression tests with individual variants. We used 33, 8, 12, and 4 risk allele SNPs which were nominally significant in BMI, WC$_{BMIadj}$, WHR$_{BMIadj}$, and BF% GWAS results in JHS, to characterize individual variants' association with BF%, SAT, VAT, and VSR ratio, respectively; each SNP represents an independent locus within the genome (based on linkage disequilibrium).

On the heatmap (S1 Fig), the primary clustering of adiposity pattern variables on the x-axis separate different types of fat. The primary clustering of the SNPs on the y-axis separates group of SNPs that are associated with increase in BF%, BMI, WC$_{BMIadj}$, and WHR$_{BMIadj}$. A clear majority of risk alleles associated with increased BMI and BF% are positively associated with SAT and overall BF%. Among WHR$_{BMIadj}$ and WC$_{BMIadj}$-increasing alleles, no discernible or consistent pattern can be observed (S1 Fig).

## Discussion

In this study we characterized the associations between genetic risk scores for BMI, WC$_{BMIadj}$, WHR$_{BMIadj}$, and BF% with body fat composition phenotypes, including overall BF%, VAT, SAT, and VSR among AA individuals in the JHS cohort.

Phenotypically, pairwise correlations among BMI, WC, BF% and SAT were strong, but considerably weaker correlations with VAT were observed (S4 Table). Although the pairwise correlation between BMI and WHR was somewhat weaker (ρ = 0.16) than observed in most studies (which often report a correlation of ~0.4 or more) [36–38], similarly weak correlations have been previously reported [39, 40], including in studies of African Americans individuals [41]. The BMI-WHR correlation is known to vary across age, sex, and ethnic groups [38].

All PRS were significant and positive predictors of BF% in AA; this illustrates that similar to most BMI-linked SNPs [16], most known BF%-associated loci, initially observed in EA individuals [19],, are likely transferrable to AA populations (e.g. associatons may be replicable across populations). Furthermore, BMI-associated loci have cross-phenotypic effects on body fat [19]; the significant association between BMI-PRS and BF% measures in our analysis may also be indicative of transferrability of similar biological mechanisms for these GWAS identified variants from EA to AA populations, though the causal SNPs at most of these loci are unknown.

Both WC$_{BMIadj}$ and WHR$_{BMIadj}$-PRS were also found to be positive predictors of BF% when associations were not adjusted for BMI; only WHR$_{BMIadj}$-PRS remained positively

associated with BF% after adjusting for BMI, albeit at an attenuated scale (Table 2). Similarly, both $WC_{BMIadj}$ and $WHR_{BMIadj}$-PRS were positive predictors of SAT, but only $WC_{BMIadj}$-PRS remained associated after adjustment for BMI. Similarly inconsistent patterns were also observed at the individual SNP level.

Examination of individual SNPs that exhibited nominally significant associations to BF%, BMI, $WC_{BMIadj}$, and $WHR_{BMIadj}$ in this cohort shows that larger proportions of BMI- and BF%- associated SNPs cluster with BF% and subcutaneous adiposity measures. $WC_{BMIadj}$- associated SNPs, in contrast, do not appreciably cluster with any of the adiposity traits.

Collectively, results from both PRS models and supplementary cluster analyses suggest that BF% and BMI SNPs seem to better predict SAT, whereas the more deleterious VAT (and VSR) are more likely represented by WHR-associated variants, of the more commonly measured adiposity indices. Difference in patterns of visceral adiposity in AA [42], with significantly lower levels of VAT in BMI-adjusted analysis noted previously [13], that suggests a favorable visceral adiposity profile compared to EA [43]. Results highlighted the importance of population-specific GWAS studies on central adiposity measures in addition to overall adiposity phenotypes like BMI. Due to the limited availability of GWAS on VAT or VSR phenotypes [44], with none in a well-powered AA cohort, we are unable to asssess whether these observed differences are primarily genetically driven. Larger studies in AA populations are needed to assess the genetics of these traits across ancestries.

This is the first study to examine the relationship between adiposity PRS, including BF%, $WC_{BMIadj}$ and $WHR_{BMIadj}$, and adiposity traits in AA. Utilization of several PRS with different risk set configurations made it possible to examine associations under comparative scenarios, minimizing potential sources of bias in estimates.

Relatively few African American cohorts with comprehensive assessment of adiposity metrics (including fat mass and CT based adiposity metrics) and genetic data exist, limiting our statistical power for these analyses. However, JHS is to our knowledge the largest such cohort and consistency of results under multiple approaches suggested a measure of robustness for estimates.

Results of this assessment pertain only to adults. Many genetic variants associated with obesity traits are age-dependent [45]. For example, in one prior study, there was no association between BMI-PRS and BF% in <5 years olds [46]; thus, extrapolation of results from this study to younger age categories in AA individuals may be unwarranted. This limitation is likely to be extended to $WC_{BMIadj}$- and $WHR_{BMIadj}$- associated SNPs since they also exhibit interaction with age [47]. We could not account for differential associations of dimorphic or sex-specific variants because of smaller numbers of male participants. Bioimpedance methods also have limitations [48], where the method slightly underestimates BF% in highly obese males [49]. However, we do not expect this limitation to substantially bias our results since only 6% of males had BMI measures >40 in this dataset.

Due to limited sample size for deriving adequate training and testing datasets, we did not perform extensive comparisons of different PRS derivation methods in JHS (for example comparisons of LDpred and pruning [22] and thresholding methods). Our unweighted risk score method, including an LD-pruned list of previously reported genome-wide significant variants in PRS derivation for each trait, may not provide ideal predictive power. However, this simple unweighted method is also likely to be less strongly influenced by differential LD and variant effect sizes across ancestral populations. Additionally, this approach allowed us to utilize SNPs both from large European focused GWAS for these traits and smaller studies in African American populations. PRS derivation and weighting by previously reported effect sizes is complex in admixed African American populations [50, 51], especially given the lack of representation of participants of diverse genetic ancestry in prior GWAS [52, 53]. Comparisons of different

methods and use of ancestry-specific effect size estimates for PRS should be explored in future work for adiposity traits. Finally, the cross-sectional nature of the assessment restricts causal inferences.

In conclusion, these analyses illustrate that anthropometric phenotype-associated loci, initially explored in predominantly EA populations, are generally transferrable to AAs. Our results suggest that total gain in fat mass in AA, at least for gains in fat mass mediated by genetic factors, may be mostly through subcutaneous rather than visceral adiposity, but a comparable assessment in other populations, including Europeans, would be required to make firm conclusions about any population differences. Absence of association between anthropometric PRS, particularly $WC_{BMIadj}$ and $WHR_{BMIadj}$, and adiposity traits like BF% may imply that the latter phenotypic measure are likely driven by genetic variants which influence overall adiposity versus central obesity.

## Supporting information

**S1 Table. SNPs used for configuration of polygenic risk scores.**
(XLSX)

**S2 Table. Number of SNPs configurations used for calculation of polygenic risk scores under complementary approaches.**
(DOCX)

**S3 Table. Polygenic risk score validation.** Estimates for all phenotypes other than percentage body fat (%BF).
(DOCX)

**S4 Table. Matrix of correlation between observed phenotypic measures for Jackson Heart Study.** Spearman correlation coefficients were calculated.
(DOCX)

**S5 Table. Coefficients of determination ($R^2$) for PRS.** They exhibit the proportion of adiposity traits'(BF%, SAT, VAT, VSR) variances were predicted by each individual PRS.
(DOCX)

**S6 Table. Associations between phenotype-PRS (columns), and measures of adiposity (rows).** Betas are reported for standardized inverse normalized values, followed by their respective p-values. Nominally statistically significant results ($p < 5.00 \times 10^{-2}$) are in bold font.
(DOCX)

**S1 Fig. Heatmap plot of association between top phenotype-linked SNPs and adiposity traits.** These SNPs were used for PRS calculation under approach 3 and represent polymorphisms with evidence of directionally consistent and statistically significant associations with their respective traits in JHS-genome wide assessment. Ward's method used for SNP and trait clustering.
(TIF)

## Acknowledgments

**Disclaimer:** The views expressed in this manuscript are those of the authors and do not necessarily represent the views of the National Heart, Lung, and Blood Institute; the National Institutes of Health; or the U.S. Department of Health and Human Services.

## Author Contributions

**Conceptualization:** Mohammad Y. Anwar, Laura M. Raffield, Leslie A. Lange, Kira C. Taylor.

**Data curation:** Mohammad Y. Anwar, Laura M. Raffield, Leslie A. Lange, Kira C. Taylor.

**Formal analysis:** Mohammad Y. Anwar.

**Investigation:** Laura M. Raffield.

**Methodology:** Mohammad Y. Anwar, Kira C. Taylor.

**Project administration:** Mohammad Y. Anwar.

**Resources:** Adolfo Correa, Kira C. Taylor.

**Software:** Mohammad Y. Anwar.

**Supervision:** Kira C. Taylor.

**Validation:** Leslie A. Lange, Adolfo Correa.

**Visualization:** Mohammad Y. Anwar.

**Writing – original draft:** Mohammad Y. Anwar.

**Writing – review & editing:** Laura M. Raffield, Leslie A. Lange, Adolfo Correa, Kira C. Taylor.

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
