## [Decision Letter · Decision Letter 0]

19 Apr 2021

PONE-D-21-03239

Genetic Underpinnings of Regional Adiposity Distribution in African Americans: Assessments from the Jackson Heart Study

PLOS ONE

Dear Dr. Anwar,

Thank you for submitting your manuscript to PLOS ONE. After careful consideration, we feel that it has merit but does not fully meet PLOS ONE’s publication criteria as it currently stands. Therefore, we invite you to submit a revised version of the manuscript that addresses the points raised during the review process.

We look forward to receiving your revised manuscript.

Kind regards,

Heming Wang, PhD

Academic Editor

PLOS ONE

Journal Requirements:

2) Please include captions for your Supporting Information files at the end of your manuscript, and update any in-text citations to match accordingly. Please see our Supporting Information guidelines for more information: http://journals.plos.org/plosone/s/supporting-information

3) Thank you for including your ethics statement: "The study protocol was approved by the participating JHS institutions_–including the Tougaloo College, the Jackson State University and the University of Mississippi Medical Center."   

4) Please provide additional details regarding participant consent. In the ethics statement in the Methods and online submission information, please ensure that you have specified whether consent was informed. If your study included minors, state whether you obtained consent from parents or guardians.

5) In your Methods section, please provide additional information about the participant recruitment method and the demographic details of your participants. Please ensure you have provided sufficient details to replicate the analyses such as: a) a table of relevant demographic details, b) a description of how participants were recruited, and c) descriptions of where participants were recruited and where the research took place.

6) Please provide a sample size and power calculation in the Methods, or discuss the reasons for not performing one before study initiation.

7)  In your Data Availability statement, you have not specified where the minimal data set underlying the results described in your manuscript can be found. PLOS defines a study's minimal data set as the underlying data used to reach the conclusions drawn in the manuscript and any additional data required to replicate the reported study findings in their entirety. All PLOS journals require that the minimal data set be made fully available. For more information about our data policy, please see http://journals.plos.org/plosone/s/data-availability.

8) Please include your tables as part of your main manuscript and remove the individual files. Please note that supplementary tables (should remain/ be uploaded) as separate "supporting information" files.

Reviewers' comments:

Reviewer's Responses to Questions

**Comments to the Author**

1. Is the manuscript technically sound, and do the data support the conclusions?

Reviewer #1: Partly

2. Has the statistical analysis been performed appropriately and rigorously? 

Reviewer #1: Yes

3. Have the authors made all data underlying the findings in their manuscript fully available?

Reviewer #1: Yes

4. Is the manuscript presented in an intelligible fashion and written in standard English?

Reviewer #1: Yes

5. Review Comments to the Author

Reviewer #1: Anwar et al use genetic variants associated with anthropometric traits in Europeans and explore their associations with several adiposity measures in African Americans using genetic risk scores.

Comments:

The correlation between WHR and BMI seems weaker than what is expected, is there an explanation from the authors to that regards? Is that specific to this population?

PRS calculation:

The authors need to provide more details about the PRS calculation in the “Methods” section. For example, what tool was used to generate it? How many variants were included? What was the explained variance for each PRS-trait. Was the JHS data split into training and target data, or not?

Clustering and Supplementary figure 1:

What did the authors use to cluster the SNPs and the traits? Did they use Betas? I don’t think that was mentioned in the methods or results sections.

The numbers in the boxes are not in agreement with the color key. For example, some red boxes are positive instead of being negative as expected based on the key. Which ones are the correct values, the color of the box or the numbers written on them? These need to be cross-checked and updated.

The authors state that: “The primary clustering of the SNPs on the y-axis separates group of SNPs…”, however, the heatmap shows only dendograms pertaining to the rows, i.e to the traits. Did the clustering algorithm cluster both SNPs and traits, or only traits? If the SNPs were also clustered (which I believe is what the authors want), could the dendograms for the columns be shown?

Supplementary Table 6:

Is there any reason why more than one GWAS was used to use its summary stats for the PRS calculation? For example, 4 studies were listed to be used for BFP. Typically, one chooses the largest GWAS to date to use in the PRS calculation. Also, is it correctly stated that this study was used, “Genotype-by-environment interactions inferred from genetic effects on phenotypic variability in the UK Biobank” . Same question regarding this study: “Shared genetic and experimental links between obesity-related traits and asthma subtypes in UK Biobank” which was used for both WHR and body fat percentage PRS creation according to Supplementary table 6 and so on.

It is surprising to use these specific studies. Also, this study (30239722) was used to generate the BMI PRS but is a study of WHR... It is not clear how and why these studies were chosen, and whether there is a specific reason behind choosing more than one for each trait, which is not typical. The PRSs are better constructed each from one study.

Minor comments:

-The last sentence in the introduction: “We also characterized … adiposity measures” this statement is not clear.

-Table 1 : Values for WHR and Body fat percentage are not clear in the table.

-I do not think Supplementary tables 2 and 6 were mentioned in the text.

-The paper needs to be revised for minor English errors, for example: line 280 : “Taken together…”; line 284: “Results highlight importance ...BMI” and several others.

6. PLOS authors have the option to publish the peer review history of their article (what does this mean?). If published, this will include your full peer review and any attached files.

Reviewer #1: No

---

## [Author Response · Author response to Decision Letter 0]

18 Jun 2021

Note : 

First reply to editor

Then reply to reviewer’s comments

Reply to Editorial:

Journal Requirements:

Answer: 

The revised manuscript has been reformatted to accord with guidelines referred above.

2) Please include captions for your Supporting Information files at the end of your manuscript, and update any in-text citations to match accordingly. Please see our Supporting Information guidelines for more information: http://journals.plos.org/plosone/s/supporting-information

Answer:

We refer you to our reply to comment #1.

3) Thank you for including your ethics statement: "The study protocol was approved by the participating JHS institutions_–including the Tougaloo College, the Jackson State University and the University of Mississippi Medical Center." 

Answer:

The method section was amended accordingly (see lines 129-132).

4) Please provide additional details regarding participant consent. In the ethics statement in the Methods and online submission information, please ensure that you have specified whether consent was informed. If your study included minors, state whether you obtained consent from parents or guardians.

Answer:

We refer you to our reply to comment #3. All JHS participants were >=21 years of age. Consent was informed. 

5) In your Methods section, please provide additional information about the participant recruitment method and the demographic details of your participants. Please ensure you have provided sufficient details to replicate the analyses such as: a) a table of relevant demographic details, b) a description of how participants were recruited, and c) descriptions of where participants were recruited and where the research took place.

Answer:

In the method section, study population sub-section was revised and expanded to include enlisted details. Please see lines 113-122.

6) Please provide a sample size and power calculation in the Methods, or discuss the reasons for not performing one before study initiation.

Answer:

Given the paucity of research on fat mass and CT based adiposity traits in African Americans, we felt study was highly important and warranted to be performed even as the power was limited by availability of the small number of African American cohorts with needed genetic and phenotypic observations. We therefore did not include power calculation. However, to minimize bias in results, we performed analyses under comparative approaches, and results suggested robustness of estimates. The discussion section was amended to highlight this issue (lines 344-347). 

You may also see our answer to comment#2 (under title PRS calculation) to reviewer #1, and see lines 358-370.

7) In your Data Availability statement, you have not specified where the minimal data set underlying the results described in your manuscript can be found. PLOS defines a study's minimal data set as the underlying data used to reach the conclusions drawn in the manuscript and any additional data required to replicate the reported study findings in their entirety. All PLOS journals require that the minimal data set be made fully available. For more information about our data policy, please see http://journals.plos.org/plosone/s/data-availability.

Answer:

A new “Data Availability” part was added before reference list, and detailed information was included under the section (see lines 396-402).

8) Please include your tables as part of your main manuscript and remove the individual files. Please note that supplementary tables (should remain/ be uploaded) as separate "supporting information" files.

Answer:

Tables were duly embedded in manuscript as per guidelines, and supplementary materials have been submitted separately. Thank you for providing directions.

Reply to reviewer’s comments: 

Reviewer #1: Anwar et al use genetic variants associated with anthropometric traits in Europeans and explore their associations with several adiposity measures in African Americans using genetic risk scores.

Comments:

The correlation between WHR and BMI seems weaker than what is expected, is there an explanation from the authors to that regards? Is that specific to this population?

Answer: 

Although the correlation in JHS is weaker than reported in most studies, similar or even lower values were previously reported in some studies, including in studies of African American individuals (perhaps as low as r=0.05 (PMID: 21487399).

We added a paragraph (lines 306-310) to the discussion section to highlight the findings.

PRS calculation:

The authors need to provide more details about the PRS calculation in the “Methods” section. For example, what tool was used to generate it? How many variants were included? What was the explained variance for each PRS-trait. Was the JHS data split into training and target data, or not?

Answer: 

Due to limited sample size for deriving adequate training and testing datasets, the study lacked adequate power to perform extensive comparisons of different PRS derivation methods, for example comparisons of LDpred and pruning [PMID: 26430803] and thresholding methods. We therefore opted to use a simpler unweighted risk score method, including an LD pruned list of previously reported genome-wide significant variants in PRS derivation for each trait. We acknowledge that this approach may not provide ideal predictive power; however, the unweighted method is also likely to be less strongly influenced by differential LD and variant effect sizes across ancestral populations. This approach allowed us to utilize SNPs both from large European focused GWAS for these traits (e.g. PMID: 31669095, 28448500 etc.) and smaller studies in African American populations (e.g. PMID:28430825). PRS derivation and weighting by previously reported effect sizes is complex in admixed African American populations, especially given the lack of representation of participants of diverse genetic ancestry in prior GWAS. Use of ancestry specific effect sizes for variants included in PRS should be explored in future for adiposity traits. 

We added a paragraph to the discussion section to highlight this methodological limitation and highlight the strength of our approach which harbors an inherently lower probability of bias (lines 358-370). 

We also added an additional supplementary table for coefficient of determination of estimates (as new supplementary Table 5) to exhibit the proportion of variance in adiposity traits explained by each phenotype-PRS. 

And finally, a short paragraph was added at the end of Methods section to correctly cite platforms used for PRS generation and heatmap plotting.

Clustering and Supplementary figure 1:

What did the authors use to cluster the SNPs and the traits? Did they use Betas? I don’t think that was mentioned in the methods or results sections.

The numbers in the boxes are not in agreement with the color key. For example, some red boxes are positive instead of being negative as expected based on the key. Which ones are the correct values, the color of the box or the numbers written on them? These need to be cross-checked and updated.

The authors state that: “The primary clustering of the SNPs on the y-axis separates group of SNPs…”, however, the heatmap shows only dendograms pertaining to the rows, i.e to the traits. Did the clustering algorithm cluster both SNPs and traits, or only traits? If the SNPs were also clustered (which I believe is what the authors want), could the dendograms for the columns be shown?

Answer:

Heatmaps were reconstructed to accommodate both traits and SNPs clustering with row and column dendograms added. We also slightly revised the method section and the caption of the supplementary figure to reflect the choice of betas for heatmap plotting. Thank you for suggestions. 

Supplementary Table 6:

Is there any reason why more than one GWAS was used to use its summary stats for the PRS calculation? For example, 4 studies were listed to be used for BFP. Typically, one chooses the largest GWAS to date to use in the PRS calculation. Also, is it correctly stated that this study was used, “Genotype-by-environment interactions inferred from genetic effects on phenotypic variability in the UK Biobank” . Same question regarding this study: “Shared genetic and experimental links between obesity-related traits and asthma subtypes in UK Biobank” which was used for both WHR and body fat percentage PRS creation according to Supplementary table 6 and so on.

It is surprising to use these specific studies. Also, this study (30239722) was used to generate the BMI PRS but is a study of WHR... It is not clear how and why these studies were chosen, and whether there is a specific reason behind choosing more than one for each trait, which is not typical. The PRSs are better constructed each from one study.

Answer: 

The reviewer has raised an important methodological consideration. While it is certainly possible to generate PRS using a single large-scale meta-analysis containing hundreds of GWAS-significant SNPs for frequently studied phenotypes like BMI, that approach would have left us underpowered for generating body fat mass PRS. The number of GWAS studies about body fat mass is limited, and none report a sizeable number of SNPs. Using multiple studies allowed us to capture a larger number of loci, and we confirmed that variants included in our final PRS scores were not in strong LD in relevant 1000 Genomes populations. We also thought it was important to accommodate lead findings from African American specific GWAS studies (e.g. PMID: 23966867 or PMID: 28430825), which are likely not well-captured by the largest existing GWAS for adiposity traits, all of which are predominantly in European ancestry populations. 

Therefore, we opted for a database approach where we extracted all GWAS-significant SNPs associated with each target trait from relevant sources, primarily GWAS-catalogue. Given that several studies, including those referred to by the reviewer, included GWAS results for several traits, it is expected that they contribute SNPs to multiple PRS. 

Additionally, we re-evaluated the list of SNPs used for PRS generation to ensure they were indeed reported by purported studies, particularly those referred above. This was done by checking GWAS-catalogue, dbSNP, and going back to original studies. The rigorous process found handful number of SNPs mis-allocated to different phenotype in the original list. Estimates in tables were subsequently updated to reflect corrections made in polygenic risk sets. 

Supplementary Table 1 was also amended, and now includes all the SNPs used for polygenic risk scores calculations under approach 1 (e.g. the principle approach used in the study).

Minor comments:

-The last sentence in the introduction: “We also characterized … adiposity measures” this statement is not clear.

Answer:

Changed to: “We also estimated the associations….”.

-Table 1 : Values for WHR and Body fat percentage are not clear in the table.

Answer:

Reporting units were amended.

-I do not think Supplementary tables 2 and 6 were mentioned in the text.

Answer:

Supplementary table 2 and 6 were revised and re-arranged as S3 and S1 tables, respectively.

-The paper needs to be revised for minor English errors, for example: line 280 : “Taken together…”; line 284: “Results highlight importance ...BMI” and several others.

Answer:

Manuscript reviewed again for grammatical mistakes.

6. PLOS authors have the option to publish the peer review history of their article (what does this mean?). If published, this will include your full peer review and any attached files.

Do you want your identity to be public for this peer review? For information about this choice, including consent withdrawal, please see our Privacy Policy.

Reviewer #1: No

---

## [Decision Letter · Decision Letter 1]

21 Jul 2021

Genetic underpinnings of regional adiposity distribution in African Americans: Assessments from the Jackson Heart Study

PONE-D-21-03239R1

Dear Dr. Anwar,

We’re pleased to inform you that your manuscript has been judged scientifically suitable for publication and will be formally accepted for publication once it meets all outstanding technical requirements.

Kind regards,

Heming Wang, PhD

Academic Editor

PLOS ONE

Additional Editor Comments (optional):

Reviewers' comments:

Reviewer's Responses to Questions

**Comments to the Author**

1. If the authors have adequately addressed your comments raised in a previous round of review and you feel that this manuscript is now acceptable for publication, you may indicate that here to bypass the “Comments to the Author” section, enter your conflict of interest statement in the “Confidential to Editor” section, and submit your "Accept" recommendation.

Reviewer #1: All comments have been addressed

2. Is the manuscript technically sound, and do the data support the conclusions?

Reviewer #1: Yes

3. Has the statistical analysis been performed appropriately and rigorously? 

Reviewer #1: Yes

4. Have the authors made all data underlying the findings in their manuscript fully available?

Reviewer #1: Yes

5. Is the manuscript presented in an intelligible fashion and written in standard English?

Reviewer #1: Yes

6. Review Comments to the Author

Reviewer #1: The authors have improved the manuscript compared to their original submission and have addressed all comments. No further comments.

7. PLOS authors have the option to publish the peer review history of their article (what does this mean?). If published, this will include your full peer review and any attached files.

Reviewer #1: No

---

## [Editor Report · Acceptance letter]

26 Jul 2021

PONE-D-21-03239R1 

Genetic underpinnings of regional adiposity distribution in African Americans: Assessments from the Jackson Heart Study 

Dear Dr. Anwar:

I'm pleased to inform you that your manuscript has been deemed suitable for publication in PLOS ONE. Congratulations! Your manuscript is now with our production department. 

Kind regards, 

on behalf of

Dr. Heming Wang 

Academic Editor

PLOS ONE